# Caffeine: The Story beyond Oxygen-Induced Lung and Brain Injury in Neonatal Animal Models—A Narrative Review

**DOI:** 10.3390/antiox13091076

**Published:** 2024-09-03

**Authors:** Stefanie Endesfelder

**Affiliations:** Department of Neonatology, Charité—Universitätsmedizin Berlin, Augustenburger Platz 1, 13353 Berlin, Germany; stefanie.endesfelder@charite.de; Tel.: +49-030-450-559-548

**Keywords:** hyperoxia, hypoxia, hypoxia–ischemia, intermittent hypoxia, oxidative stress, caffeine, newborn rodent, developing brain, immature lung

## Abstract

Caffeine is one of the most commonly used drugs in intensive care to stimulate the respiratory control mechanisms of very preterm infants. Respiratory instability, due to the degree of immaturity at birth, results in apnea of prematurity (AOP), hyperoxic, hypoxic, and intermittent hypoxic episodes. Oxidative stress cannot be avoided as a direct reaction and leads to neurological developmental deficits and even a higher prevalence of respiratory diseases in the further development of premature infants. Due to the proven antioxidant effect of caffeine in early use, largely protective effects on clinical outcomes can be observed. This is also impressively observed in experimental studies of caffeine application in oxidative stress-adapted rodent models of damage to the developing brain and lungs. However, caffeine shows undesirable effects outside these oxygen toxicity injury models. This review shows the effects of caffeine in hyperoxic, hypoxic/hypoxic-ischemic, and intermittent hypoxic rodent injury models, but also the negative effects on the rodent organism when caffeine is administered without exogenous oxidative stress. The narrative analysis of caffeine benefits in cerebral and pulmonary preterm infant models supports protective caffeine use but should be given critical consideration when considering caffeine treatment beyond the recommended corrected gestational age.

## 1. Introduction

### 1.1. Caffeine in Preterm Infants

The use of caffeine in premature infants is undisputed. In the late 1970s, Aranda et al. [1] already described the reduction of AOP frequency in preterm infants aged 28 weeks on average. The physiologically influenced effects of caffeine are due to the stimulation of the respiratory control centers in the medulla by competitive inhibition of adenosine and increased contractility of the diaphragm [2,3]. The fact that caffeine is still the first drug of choice for the treatment of AOP is justified by a large number of clinical studies with a high level of evidence, beginning with the CAP studies by Schmidt et al. [4] on the proven benefits of caffeine in the treatment of premature infants [5]. Besides reduced apnea frequency, the most important caffeine effects are reduced incidences of BPD, chronic lung disease, death, and cerebral palsy [4,6,7,8,9]. By adolescence, the caffeine effect is lost in preterm infants, but caffeine-treated children still benefit from improvements in motor function and visual perception at the ages of 5 and 11 years [10,11,12,13]. With lower levels of evidence, caffeine appears to reduce postoperative apnea and induce anti-inflammatory cytokine profiles [14]. An ongoing debate on the timing of the initiation of caffeine therapy for preterm infants seems to confirm an early start within the first three days of life [15,16,17]. Individual clinical studies repeatedly report the harmful effects of caffeine in premature newborns. These include reduced cerebral oxygenation and decreased cerebral and intestinal blood flow velocity [18,19], increased blood pressure and tachycardia [20], a pro-inflammatory cytokine profile [21], as well as an increase in seizure frequency and seizure burden [22]. In the current context, Parladori et al. [23] revealed an improvement in cerebrovascular reactivity and pointed to the safety of caffeine in a hemodynamically vulnerable phase with positive effects on cerebral hemodynamics.

### 1.2. Pharmacology of Caffeine

Caffeine is hydrophilic and lipid-soluble, can therefore be easily distributed in all tissues and body fluids, and is membrane-permeable [3]. However, there are differences between the metabolism of newborns and adults when caffeine is catalyzed in the liver. The reason for this is the immature enzyme production of CYP1A2 in the liver of newborn and premature infants. CYP1A2, also known as cytochrome P450, catalyzes caffeine primarily into paraxanthine, theobromine, and theophylline [24]. During the initial months of life, caffeine is excreted renally, in an almost unmetabolized form, perhaps due to lower, ontogeny-related CYP1A2 activity. As a result, the metabolic products tend to play a subordinate role in potential caffeine effects until the newborn metabolism is fully available at approximately one year of age. This is also accompanied by a slower metabolism and longer serum half-life of caffeine [3,25,26,27]. These processes can be influenced by the immaturity of the organism, body weight, and necessary nutritional strategies [28,29,30].

The pharmacological effects of caffeine in clinically relevant doses for the premature infant are mediated, on the one hand, via non-specific antagonism at the adenosine receptor (AdoR) subtypes AdoRA1, AdoRA2a, and AdoRA2b receptors [31,32,33,34,35]. Here, genetic variability due to polymorphisms is not negligible and can modulate the effects of caffeine [36]. On the other hand, experimental studies also point to an important role of caffeine via antioxidant properties [37,38,39,40], with high relevance to the fact that premature infants have an immature antioxidant enzyme system [41]. This is accompanied by anti-inflammatory and cell death inhibitory effects, with communication via relevant cellular signaling pathways playing a more important role in mediating the protective effects of caffeine. This aspect is presented in the discussion based on the data collected in this review article.

### 1.3. Postnatal Oxidative Stress Injury Models for the Immature Lung and Developing Brain

Oxidative stress, as a fundamental risk factor for initially induced diseases of prematurity, is not new and is referred to as oxygen-related diseases of prematurity [42]. Premature infants are highly vulnerable to oxidative stress, which is already unavoidable due to premature birth [43,44]. Early gestational age and low birth weight are negatively correlated with an adequate cellular response to ROS [45], which can be increased by respiratory instability and oxygen therapies being initiated [42,46]. An essential tool for managing the consequences of oxygen adaptation and thus oxidative stress is to observe the oxygen saturation of the premature infant and make sensitive adjustments as required [47]. Despite reliable findings on therapy-optimized oxygen saturations, further studies with reliable outcome results are still required.

Looking at the pulmonary development and the maturation of the brain, which is orchestrated in the intrauterine environment under hypoxic conditions, it is comprehensible and provable that a too-early change due to a too-early birth into an extrauterine hyperoxic environment is associated with oxidative stress. Especially in the last trimester of pregnancy, neuronal processes such as cell division, differentiation, migration, synaptogenesis, myelination, plasticity, and physiological cell death are very vulnerable to non-physiological noxae. The same applies to pulmonary development, which, in prematurely born children, must first develop from the saccular phase into the alveolar mature phase. Oxidative stress causes impairments in the structure and, thus, in the function of both immature organ systems due to multiple mechanisms [48,49]. Some of these causes consequential damage into adulthood. This may increase the risk of adults who survive a premature birth developing chronic diseases, including respiratory disorders such as asthma or COPD, as well as neurological and psychiatric disorders such as ADHD or autism spectrum disorder [50,51]. It should be noted that the more immature newborns are the more susceptible they are to redox-sensitive situations. The risk of developing short- and long-term neurological complications is not only based on the consequences of oxidative stress on the premature brain. It is now known that children with severe BPD, in addition to pulmonary dysfunction in adolescence, can also be neurologically impaired in the long term without additional brain injury [52,53,54].

#### 1.3.1. Oxidative Stress in Premature

There are various scenarios in which oxidative stress can arise and have an impact on a premature situation. Hyperoxia (HY) per se is unavoidable. The change from the hypoxic environment of the fetus (25–30 mmHg), which is also necessary for normal development, to a hyperoxic environment during birth (75–85 mmHg), generates relative hyperoxia [55], which leads to oxidative stress in an underdeveloped antioxidant system and a lack of alveolar surfactant. Respiratory instabilities sometimes require medically induced oxygen supplementation, which then leads to absolute hyperoxia. In addition, inadequate respiratory control, with apneas and impaired inspiratory efforts, are major triggers for intermittent hypoxia (IH) events [56]. There are few data on the possible long-term consequences of postnatal IH. Experimental and clinical data suggest the induction of pathological cascades via an oxidative stress-induced signaling pathway, with subsequent inflammation [57,58]. Besides asphyxia, hypoxia (HO) or hypoxia/ischemia (HI) in the vulnerable phase of premature infants is one of the most common causes of death and morbidity [59,60]. There is a generalized higher risk of hypoxia events the more immature the organism is at birth, and this is compounded by fetal immobilization with slowed metabolism and reduced respiratory drive during the transition from hypoxic to hyperoxic environments [60]. Hypoxia and hypoxia/ischemia cause oxidative stress and are, in turn, involved in the pathogenesis of neurological diseases such as hypoxic–ischemic encephalopathy [61]. Neuroinflammation and mitochondrial dysfunction alter the cellular status, resulting in irreversible damage [46,62].

#### 1.3.2. Postnatal Oxidative Stress Injury Models

The adequate injury models used for experimental research are based on the transferred clinical oxygen conditions from the neonatal situation. The rodent hyperoxia (HY) injury model is largely based on FiO_2_ 80% oxygen, which is clinically avoided in neonatal situations. The natural fetal situation during the last trimester of pregnancy is physiologically consistent with a low oxygen concentration until parturition in room air causes an increase in oxygen partial pressure from 25–30 mmHg to 75–85 mmHg [60]. The exogenous rodent hyperoxia model provides a 4-fold increase in oxygen levels and causes an increase in in vivo oxygen partial pressure of 55–165 mmHg in newborn mice [63]. Therefore, 80% oxygen exposure mimics a large increase in oxygen concentration in a neonate with a very immature stage of brain and lung development. The plausibility of the model is based on demonstrably comparable neurological and pulmonary pathologies in premature infants with extremely low and very low birth weights [38,39,48,63,64,65,66,67,68,69,70,71,72,73,74,75].

Hypoxia (HO), with oxygen as the only variable, tends to be a criterion for reaching the pathological state in respiratory-associated rodent models. In experimental hypoxic studies in neonatology research, hypoxia is used to represent oxygen deficiency per se [76,77,78]. Triggering factors in the clinical situation for hypoxia or transient hypoxia are respiratory instabilities, e.g., apnea, infections of the upper respiratory tract, as well as hypoxia under hyperoxia during ventilation of the immature premature infant. Chronic hypoxia in newborn rodents often serves as a model of BPD-induced brain damage [79] and is simulated with oxygen concentrations of usually 8–10% oxygen, whereby the duration of exposure is highly variable. In the case of alternating oxygen concentrations in FiO_2_ between normoxia (21% O_2_) and hypoxia, the term intermittent hypoxia (IH) is used. In the neonatal clinical setting, the PaO_2_ drops to at least 80% [58]. This is also caused by underdeveloped respiratory control and limited lung capacity, which can be exacerbated by BPD. The experimental models are quite variable, with intermittent oxygen concentrations between normoxia and hypoxia (21%/ 5% O_2_) or moderate hyperoxia with hypoxia (50%/5–12% O_2_) [80,81,82].

Hypoxia and intermittent hypoxia-injury models usually cause high variability. Using the Rice–Vannuccie model [83,84], i.e., hypoxia after an ischemic insult (HI), this can be manifested. The model is based on a unilateral ligation of the carotid artery with subsequent recovery of the pups under the care of the dam (~1–2 h), only to be exposed to hypoxia. The pups remain under hypoxia (8–10% O_2_) for 1–3 h. Subsequent comparisons are made between the damaged hemisphere and the contralateral hemisphere exposed to hypoxia without ischemia. Clinically, this model represents perinatal asphyxia and hypoxic–ischemic encephalopathy (HIE).

All of these damage models related to oxidative stress reflect different settings in the everyday clinical care of perinatal centers. However, they all have one thing in common: a wide range of neuronal and pulmonary damage occurs depending on the severity, duration, and organ-related developmental time of the damage. Even very short phases of oxidative stress cause degeneration, impaired proliferation, or differentiation of precursor cells. Ultimately, these impairments can essentially disrupt function and thus cause a poor long-term outcome [48,67,72,85,86,87,88,89].

This review focuses on rodent oxygen damage models due to the large amount of experimental data available. It should not go unmentioned that large animal models, such as the premature pig or sheep, can extend neuronal and pulmonary development as a premature infant model for developmental studies or investigations of other variables [90,91].

#### 1.3.3. Rodent Model for Preterm Brain Development

In neuroscience, the development of differences between humans and the maturation of the rodent brain is well known. The newborn rodent is very well suited as a model organism for the comparative neuronal development of premature infants [92,93]. Pups are often used within the first two weeks of life. This is due to the phase of rapid brain growth [94]. This phase in rodents is comparable to the developmental stage of premature infants in the 23rd week of pregnancy. At this age, the most important steps of brain development can be deciphered, which occur in both rodents and humans, and thus describe a comparable time course of postnatal brain development in all species [93,95].

#### 1.3.4. Rodent Model for Postnatal Pulmonal Maturation

Oxygen toxicity-induced pulmonary injury models [66,96] are comparable in pathology to human lung-injured preterm infants [97,98,99]. In newborn postnatal rats, the respective developmental stage at birth correlates with the saccular phase (E18–P4; human gestational age 24–38 weeks) and with the first phase of alveolarization (P4–P21; human gestational age 36 weeks to postnatal age), which overlaps with the second/continuous phase of alveolarization (P14–P60) [97,100]. Following these developmental dependencies, rodents are usually used in oxygen-related models from the day of birth until about the third week of life.

### 1.4. Objective of This Study

The aim of this review is to provide a comprehensive overview of the effects of caffeine in the recognized postnatal oxygen toxicity models and to specifically outline the effects of caffeine in non-oxygen toxicity models. This narrative analysis is intended to provide the basis for discussion of whether the benefits of caffeine in cerebral and pulmonary preterm neonatal injury models warrant caffeine treatment in the clinical setting of preterm neonatal care beyond the recommended corrected gestational age of 34 weeks gestation (postmenstrual age, PMA). 

## 2. Materials and Methods

The work included a descriptive study with a narrative review to demonstrate caffeine effects in the associated preterm oxidative stress rodent injury models of the immature lung and developing brain. A major part of the review describes caffeine effects in the absence of oxygen toxicity in the pups.

A structured search of PubMed, Scopus, and Google Scholar databases was performed within the last 20 years (2004–2024) using descriptors such as Medical Subjects Headings (MeSH) as well as entry terms to explicitly examine the title, abstract, and study design in the methods section. Likewise, reference lists were used to further search for related references in the selected articles. Individual images from BioRender.com were used to create the overview image and the graphic additions in the tables.

## 3. Results

### 3.1. Caffeine and the Hyperoxic-Injury Model

#### 3.1.1. Caffeine Protects the Immature Lungs Affected by Hyperoxia

To compare the data recorded for the use of caffeine in postnatal rodent pulmonary injury models by high oxygen concentrations (HY), eight studies [101,102,103,104,105,106,107,108] were included in this review (Table 1). The rodent species used included mice (C57Bl/6, FVB/n), rats (Sprague Dawley), and rabbits (New Zealand white).

The beginning of hyperoxia in these studies primarily starts on the day of birth (P0), except for in the study by Sadek et al. [106], which starts on postnatal day 7 (P7). The caffeine concentrations used relate to caffeine base and range from 5 to 25 mg/kg/day, whereby caffeine was administered constantly over time in the same concentration or starting with a bolus on the first day of administration. In the study by Chen et al. [101], caffeine was administered via the lactating mother. The start of caffeine therapy is either identical to the start of oxygen exposure or scheduled up to 48 h after hyperoxia.

In six [101,102,103,105,106,107] of the eight studies, a clear protective effect of caffeine on hyperoxia-damaged lungs was demonstrated. Overall, anti-oxidative, anti-inflammatory, and anti-apoptotic effects were shown, which were accompanied by an improvement in the damaged lung morphology. Some studies considered the signaling pathways involved and were able to demonstrate the modulation of redox-sensitive signaling mediations via NfκB [101] or HIF [102]. Chen et al. [101] also demonstrated increased surfactant translation. Improved microvascularization was demonstrated in the studies by Jing et al. [105], Dumpa et al. [102], and Teng and colleagues [103]. Teng et al. also proved that caffeine reduced ER stress induced by hyperoxic exposure as well as mitochondrial dysfunction [103]. Two studies were unable to show a protective effect for caffeine after hyperoxia damage [104,108], with Dayanim et al. [108] demonstrating not only pro-apoptotic and pro-inflammatory caffeine effects but also an influence on pulmonary cellular organization. The expression of AdoRA2a was modulated in two out of these eight studies [101,108]. Hyperoxia in addition to caffeine led to a demonstrable weight gain in the pups in four of the studies [101,102,104,105].

#### 3.1.2. Caffeine Protects the Developing Brain Affected by Hyperoxia

To compare the data recorded for the use of caffeine in postnatal rodent brain injury models by high oxygen concentrations (HY), two studies [109,110] were included in this review (Table 2). Rats (Sprague Dawley) were used as the rodent species in these studies.

Both studies used PaO_2_ 50% with an exposure duration of 14 days. Caffeine administration started at the same time as hyperoxia on the day of birth, and both studies started at P0 with a bolus of 20 mg/kg/day and 5 mg/kg/day thereafter. Soontarapornchai et al. [110] extended the trials to the clinically adapted dose (LoC) with high-dose caffeine (HiC) of 80 mg/kg/day and 20 mg/kg/day.

In both studies, caffeine demonstrated its neuroprotective effect via antioxidant, anti-inflammatory, and/or anti-apoptotic effects [109,110]. Batool et al. [109] showed a reduction in demyelination after hyperoxia, as did Soontarapornchai et al. [110] under high caffeine doses. Caffeine mediated more neurons, increased dendritogenesis, and reduced AdoRA1 and AdoRA2a expression at clinically relevant caffeine doses as well as at high concentrations. Adverse effects occurred at high caffeine doses, including IVH and histopathologic abnormalities. Low-dose caffeine led to weight gain under HY, while high-dose caffeine resulted in weight loss [110].

#### 3.1.3. Caffeine Protects the Lung–Brain Axis Affected by Hyperoxia

Very few research groups seem to study the effects of caffeine on the lung–brain axis in parallel. In our group, we have so far been able to study the neuronal as well as pulmonary effects of caffeine in two rat hyperoxia injury models with different experimental designs (Table 3) [38,39,68,69,70,71,75]. The same animal cohort was examined for several aspects of the analysis.

In the first study, the focus was on perinatal brain and lung injury in the presence of oxygen damage (HY) over 24 and 48 h, starting from P6 [38,69,75]. Intraperitoneally, the Wistar rat pups received 10 mg/kg once before the beginning of hyperoxia. In the second study, two periods of exposure to high oxygen were also determined [39,68,70,71]. The newborn Wistar rats were exposed to 80% O_2_ for 3 or 5 days from the day of birth and analyzed; additionally, they were supplemented with extended analyses after survival in room air of up to P15. Caffeine at 10 mg/kg was administered from P0, with plasma level monitoring, every two days.

Both studies revealed pulmonary and neuronal protective effects mediated by caffeine. Antioxidant, anti-inflammatory, and anti-apoptotic properties were demonstrated [38,39,68,75]. Caffeine prevented oxygen-mediated simplification of lung structure [75], modulated NfκB/Nrf2 signaling [39], and modulated AdoRA1 expression [68] in lung tissue. Caffeine positively influenced hyperoxia-induced impairment of both hippocampal and cerebellar neurogenesis [38,69,70,71]. The neurotrophin levels that were reduced by hyperoxia were recovered with caffeine [71].

### 3.2. Caffeine and the Hypoxic-Injury Model

#### Caffeine Protects the Developing Brain Affected by Hypoxia

The literature search for experimental studies using only low oxygen concentrations (HO) as oxidative stress triggers revealed two studies with caffeine to be included (Table 4) [111,112]. Both studies conducted neuronal experimental studies with mice, with only Back et al. [112] mentioning the animal species (C57Bl/6).

In the study by Li et al. [111], seven-day-old mice were exposed to 8% O_2_ for 20 min, with caffeine being administered from P4 with a bolus of 20 mg/kg via oral gavage, followed by 15 mg/kg (p.o.) until P7. The mouse pups in the study by Back et al. [112] received caffeine via the lactating dam from P2 for ten days (up to P12) and were exposed to low oxygen concentrations at 10% O_2_ from P3 to P12.

The analyzed endpoints of the studies showed a caffeine-prevented HIF1α accumulation in the cortex after hypoxia [111], as well as reduced ventriculomegaly and caffeine-mediated protection against demyelination [112].

### 3.3. Caffeine and the Hypoxic–Ischemic-Injury Model

#### Caffeine Protects the Developing Brain Affected by Hypoxia–Ischemia

Nine experimental in vivo studies with induced hypoxia–ischemia (HI) and caffeine application were identified, all of which focused on neuronal effects (Table 5) [113,114,115,116,117,118,119,120,121].

Initially, the rats (Wistar, Sprague Dawley) or mice (C57Bl/6) pups were always subjected to carotid artery ligation (CL), although the age varied. Four studies placed the CL closer to birth (P2, P3) [113,114,115,116], one study at P6 [118], two studies at P7 [120,121], and two studies at P10 [117,119]. The ischemia set in this way was maintained for from one to two hours, although Potter et al. [118] did not provide any specific details. The subsequent hypoxia after recovery in the mother was performed at 8%, 10%, or 5.6% O_2_ and varied between one hour and three hours (see Table 5). In four studies, caffeine was administered directly after hypoxia [117,118,119,121] and, in some cases, with additional repetitive caffeine doses up to a maximum of 24 h after hypoxia [117,118]. Kilicdag et al. [120] administered the first dose of caffeine before hypoxia and continued the applications until 72 h after hypoxia. In three studies, rat pups were treated with caffeine prior to CL, starting 24 h before the insult. Caffeine was administered well beyond HI until P6 [113,114,115]. Sun et al. [116] chose fetal premedication via the pregnant mother from embryonic day 8 (E8) before HI or starting from P3 after HI and continued caffeine treatment via the lactating mother until P16. The caffeine doses for eight of the studies mentioned were from 5 to 10 mg/kg (i.p.). There were five studies that demonstrated that caffeine had anti-oxidative, anti-inflammatory, and/or anti-apoptotic effects [113,114,115,117,120].

Hypoxia–ischemia led to impairments in cognitive and motor learning. Caffeine was able to protectively influence these deficits in the HI damage model [118,119,121]. The caffeine-mediated effects were somewhat more complex in the studies by Yang et al. [114,115] and Sun et al. [116]. Preconditioning with caffeine before the actual HI insult was able to restore the reduced synaptic proteins, alleviate the suppressed cognitive impairment, reduce ventriculomegaly, protect against demyelination, and promote the polarization of microglia to the M2 phenotype [114,115]. The fetal preconditioned rat pups in the study by Sun et al. [116] displayed a lower number of cerebral infarcts and a lower infarct volume and restored the suppressed EEG brain activity. Application of caffeine only after the HI insult showed no influence on the infarction itself, but positively influenced the suppressed EEG brain activity. A one-day caffeine premedication promoted mitochondrial functions [113].

### 3.4. Caffeine and the Intermittent-Hypoxic Injury Model

#### 3.4.1. Caffeine Protects the Immature Lung Affected by Intermittent Hypoxia

The search for pulmonary-related experimental studies with intermittent oxygen concentration (IH) as a trigger for oxidative stress resulted in two studies that considered the insult of caffeine (Table 6), both of which originated from the same working group around Jiang et al. [122,123].

Three-day-old rats (Sprague Dawley) were used in both studies. The study design was identical, but the caffeine doses and the administration route varied. Caffeine was administered either orally at 15 mg/kg [122] or intraperitoneally at 10 mg/kg [123]. Pups were transferred to intermittent hypoxia on postnatal day 3 and remained there until P12. A ten-minute cycle involving an intermittent change from 21% O_2_ to 5% O_2_ was repeated six times within one hour, followed by a one-hour break below 21% O_2_. A total of six IH cycles were performed per day.

The two analyzed caffeine concentrations via different application routes were able to successfully reduce the apnea frequencies triggered by IH per day.

#### 3.4.2. Caffeine Protects the Developing Brain Affected by Intermittent Hypoxia

Intermittent hypoxia (IH) and caffeine as an entrained substance were analyzed in two studies investigating the neuronal effects (Table 7) [109,110].

In both studies, newborn rat pups were used for the experiments, with a similar experimental design. Newborn rats (P0, Sprague Dawley) were transferred to IH shortly after birth and remained there until the end of the second week of life (P14). Pups were maintained under moderate hyperoxia (50% O_2_) and underwent three episodes of intermittent hypoxia (12% O_2_), each lasting 10 min every 2.5 h. A total of eight IH cycles were performed per day. Batool et al. [109] also started the administration of caffeine with a bolus of 20 mg/kg/day at the beginning of IH, followed by 5 mg/kg/day. Soontarapornchai et al. [110] extended the trials to the clinically adapted dose (LoC, 20 mg/kg bolus, 5 mg/kg i.p.) with a high caffeine dose (HiC) of 80 mg/kg/day (bolus) and 20 mg/kg/day (i.p.).

Batool et al. [109] reported an anti-inflammatory effect of caffeine in the IH model of the postnatal rat, which was associated with attenuation of histopathological changes in the cerebral cortex. Side effects in this study were hypermyelination of the juvenile brain, as well as an increased brain–body weight ratio. Examining the effects of caffeine in the study by Soontarapornchai et al. [110], low-dose caffeine (LoC) under IH was found to have anti-apoptotic effects associated with increased neuron numbers as well as increased dendritogenesis. Similarly, caffeine under oxidative stress had a pro-oxidant effect, increased demyelination, and led to histopathological abnormalities in the cerebral cortex. Higher concentrations (HiC) of caffeine also had an anti-apoptotic effect with increased neurons and dendritogenesis, but they also exhibited an antioxidant effect. Both concentrations were able to modulate AdoRA1a and AdoRA2a gene expression. Low caffeine dosages induced body length, while high dosages led to weight loss as well as reduced brain weight.

### 3.5. Caffeine and the Immature Lung and the Developing Brain

#### 3.5.1. Caffeine in the Immature Lung without Oxidative Stress

Eight studies [101,102,103,104,105,108,122,124] were included in the analyses of the effect of caffeine on the postnatal immature lung (Table 8). The rodent species used included mice (C57Bl/6, FVB/n) and rats (Sprague Dawley).

Since the groups without oxygen exposure are included in lung injury models and this corresponds to the developmental stage, caffeine application began on the day of birth (P0) in six studies. Caffeine administration was initiated in the study by Teng et al. [103] at P2 and, in the study by Julien et al. [122], at P3. The analysis was mainly performed after the last caffeine administration, except for in the studies by Teng et al. [103] and Dumpa et al. [102], who kept the pups without caffeine administration until the third week of life and the 12th week of life, respectively. The caffeine concentrations were in the range of from 10 mg/kg to 25 mg/kg for clinical use, except in the study by Dayanim et al. [108], which used a bolus administration of 20 mg/kg followed by 10 mg/kg, and the study by Chen et al. [101], in which the pups were supplied with caffeine via the lactating mother.

Caffeine exhibited no beneficial effects on the development of the lungs in any study, rather, the opposite effects were observed. Dayanim et al. [108] demonstrated, in newborn mice, that caffeine had a pro-apoptotic effect and also showed pro-inflammatory effects via increased infiltration with macrophages and induction of chemokine expression. Caffeine also decreased the level of AECII and reduced the transcription of surfactant protein C [108]. The translation of ER-phagy-associated proteins, which are involved in the continuous remodeling of the endoplasmic reticulum (ER) via selective autophagy [125], was induced by caffeine in the study by Teng et al. [103].

This was accompanied by an activation of the unfolded protein response (UPR), a reaction of the ER to oxidative stress [126], as well as reduced blood vessel formation. Reduced Smad2 phosphorylation was demonstrated in the study by Rath et al., which is associated with epithelial remodeling [127]. Side effects of caffeine treatment were also reduced AdoRA2a expression [101] and induction of HIF-mapped and vascular-associated transcripts [102]. The effect of caffeine on body weight was both increasing [102] and decreasing [105], whereby caffeine increased minute ventilation and respiratory rate as a side effect [122].

#### 3.5.2. Caffeine in the Developing Brain without Oxidative Stress

To obtain an overview of the effects of postnatal administered caffeine on brain development, ten studies were evaluated (Table 9) [109,110,128,129,130,131,132,133,134,135]. Nine of these studies used rats (Wistar, Sprague Dawley, and Long Evans), and one used mice (species not reported).

The caffeine application start points and duration were highly variable in these experiments. Two studies started at P0, with daily application until P14. Caffeine was dosed via a bolus (P0) of 20 mg/kg [109,110] or 80 mg/kg (i.p.) [110], followed by a constant dose of 5 mg/kg or 20 mg/kg. Kasala et al. [129] administered 50 mg/kg subcutaneously over four days, starting at P3, and analyzed the effects at P28. Two studies [131,133] had similar experimental designs, which administered caffeine at P2/P3 (20 mg/kg) and subsequently for five days (P6/P7, 15 mg/kg), whereby the routes of administration (s.c./i.p.) differed. The study by Desfrere et al. [134] also started at the same age (P3), here with mice, with a caffeine concentration of 10 mg/kg (i.p.), continuing until P10 (2.5 mg/kg (i.p.). Tchekalarova et al. [135] performed the experiments with two different caffeine doses (10 mg/kg and 20 mg/kg) at P7 with daily administration (s.c.) until P11. Caffeine was administered once to the rat pups in three studies at P6 or P7 [128,130,132], with a short observation period of 24 h (up to P7) or with additional examination times at P50/P90 [128].

Caffeine exhibited pro-oxidative [110], pro-inflammatory [109,132], pro-apoptotic [110,129,132], and/or pro-necrotic effects, resulting in neuronal disorganization, neurodegeneration [110], and mitochondrial dysfunction at the cellular level [129]. In detail, caffeine impaired neuronal proliferation [130,134], hippocampal neurogenesis [130], neuronal quantity [131], as well as astrocytogenesis [134]. In subsequent behavioral tests, the caffeine-treated animals showed hyperlocomotion, reduced social interaction, and altered anxiety behavior [128], as well as reduced motor performance and hyperactivity [135]. In addition to hypermyelination [109,110] and neurotrophin induction [130] after caffeine treatment, a modulation of AdoRA1 and AdoRA2a gene expression was also demonstrated, either with activation [132,133] or inhibition [110], accompanied by an early maturation of the adenosinergic system [133].

#### 3.5.3. Caffeine in the Lung–Brain Axis without Oxidative Stress

Based on the studies already presented by our group (see Table 10), these also allow us to observe the caffeine-treated rat pups under normoxic environmental conditions without the presence of oxidative stress as a noxious agent [38,39,68,69,70,71,75]. All studies used accompanying caffeine-treated animals, which were analyzed in parallel.

We focused the analyses on a single application (P6) of caffeine with short-term survival (P7/8) [38,69,75] and a multiple application from the day of birth (P0) to P3 or P5 [39,68,70,71], to then analyze survival without further caffeine administration up to P15 in addition to these study time points. Caffeine at 10 mg/kg (i.p.) was administered.

Both a single dose of caffeine and multiple two-day near-birth administrations mediated pro-inflammation in the postnatal lungs of rat pups [68,75], with demonstrable modulation of AdoRA1 gene expression [68]. Prolonged administration of caffeine reduced body weight for as long as caffeine was administered but normalized by the second week of life without further caffeine administration [68].

Caffeine also impaired neuronal development and induced pro-inflammatory effects. A single dose of caffeine induced proinflammatory cytokines and modulated the NfκB signaling pathway [38]. Similarly, this single administration of caffeine was sufficient to impair hippocampal neurogenesis and proliferation and prevent physiological apoptosis [69]. Similar effects can be demonstrated for repeated doses starting at P0. Here, too, hippocampal neurogenesis and proliferation capacity are reduced [71]. Giszas et al. [70] showed that neurogenesis and migration are modulated in the cerebellum, and, in addition to induction of AdoRA1, AdoRA2a, and AdoRA2b gene expression, cerebellar proliferation was upregulated.

## 4. Discussion

### 4.1. Caffeine, the ‘Gamechanger’ for Oxidative Stress Injury in the Brain–Lung Axis

Caffeine demonstrated very impressive results in the sum of all these presented studies indicating that, under oxidative stress in any model, the vulnerable, developing brain and the postnatal still immature lung in the rodent model can benefit to a high degree from the radial scavenger caffeine, and this can protect against the effects of ROS. Cerebral and cerebellar perfusion is developmentally more susceptible to reperfusion injury and oxidative stress [136]. Oxidative stress can have multiple effects on the immature organism of a premature infant [62,137]. There are molecular mechanisms that demonstrate a link between oxidative stress and the effects of caffeine. Clinical and experimental studies show this primarily through protective effects on neurological outcomes and improved lung protection [38,39,68,69,70,71,75,138,139,140,141,142,143].

The proven effects of oxygen toxicity and the effect of caffeine, combined with the damage models, are displayed schematically in Figure 1. In detail, ROS caused increased apoptosis rates, inhibited proliferation capacities, and impaired the impairment of differentiation stages in relevant progenitor cells through the oxidation of macromolecules [38,39,110]. Lipid peroxidation and peroxidative membrane lipid damage triggered by ROS or corresponding initiators lead to critical cell damage and thus cell death. In addition, this process causes inflammation [144]. DNA lesions triggered by oxidative stress were mediated via various signaling pathways and induced inflammation and apoptosis [145]. If uninterrupted, this can lead to irreversible damage and the pathogenesis of diseases of prematurity by impairing structure and function [56,138,140,146,147].

Teng et al. [103] were able to identify ER stress as a co-triggering factor for oxygen damage in a hyperoxia model of the newborn rat. Disturbances of physiological ER functions caused an accumulation of newly synthesized unfolded and misfolded proteins (UPR) [126]. The combination and responsiveness of different UPR signaling pathways after ER stress determines cell survival or death [148,149]. If the ER stress-inducing factors cannot be rebalanced, the UPRs trigger pro-apoptotic signaling pathways. ER functions, such as the maintenance of mitochondrial function and the appropriate limitation of oxidative stress, are closely related to alveologenesis [150]. ER and oxidative stress are two components that have been observed in BPD animal models [151,152]. Induction of ER stress with oxidative stress plays a non-negligible mechanistic component in injury to the developing brain [153].

Nrf2 and NfκB are important factors in an adequate response to oxidative stress and inflammation. They act independently of each other on intrinsic and extrinsic agents, but they can certainly interact [154]. Both transcription factors in oxidative stress models are influenced by the induction or inhibition of the translation of anti-oxidative, but also pro-inflammatory response molecules, and the formation of NLRP3 inflammasomes [38,39,101]. The essential function of NfκB has been described in many experimental studies, whereby the neonatal lung appears to express higher levels of protection against oxidative damage [155] and more protected alveologenesis upon sustained activation of this transcription factor [156]. The redox-sensitive transcription factor Nrf2 mediated protection against oxidative stress through transcriptional activation of various antioxidant enzymes via the antioxidant response element (ARE) [157,158].

In a clinical study, preterm infants had lower BDNF levels than term-born infants [159], with lower BDNF often correlating with reduced neurodevelopmental outcomes [160,161]. BDNF, as well as other neurotrophic factors, plays a significant role in the normal development and function of both the respiratory and nervous systems [162,163,164]. Oxidative stress and inflammation are important factors, but they can influence neurotrophins. Damage models of the immature lung and brain also demonstrate this through downregulated neurotrophic factors [70,71,165].

Epithelial damage and fibrosis are consequences of BPD [166]. Alveolar growth is controlled by angiogenesis and vascular branching [167]. Various growth factors play an essential role in this process, such as TGFβ. Dysregulated TGFβ causes a simplification of lung morphology under hyperoxia as well as hypoxia [168,169]. Vascular mediators, such as VEGF, mediate the growth and remodeling of endothelial cells in the lung [170]. The impairing effect of oxidative stress on epithelial and vascular remodeling, as well as the impairment of pulmonary alveolar formation by a simplified structure of the lung, were displayed in numerous experimental studies cited here [39,101,102,103,104,105,106,107].

Caffeine, with its pleiotropic effect, was able to effectively counteract these effects triggered by oxidative stress via different oxidative stress-associated models. As can be seen in the overviews of the individual oxygen damage models (Table 1, Table 2, Table 3, Table 4, Table 5, Table 6 and Table 7), the neuronal and pulmonary protective properties of caffeine are independent of oxygen partial pressure, duration, and generally also of the application. In connection with the proven antioxidant effects via classical biomarkers, this, again, strongly supports this property. However, it is also evident that caffeine modulates the adenosine receptors under oxidative stress so that the caffeine effect is not a direct effect via a single signaling cascade. The response of caffeine to oxygen toxicity is mediated via anti-oxidative, anti-inflammatory, and anti-apoptotic signaling cascades in the brain–lung axis [24,137].

The pulmonary effects are associated with improved microvascularization and alveologenesis [39,101,102,103,105,106,107]. Caffeine also influenced ER stress and reduced mitochondrial dysfunction, and it mediated this via fibrosis- and vascular-associated transcripts [103]. In fact, these studies showed no effect of caffeine on oxidative stress [104,108]. Caffeine, as a non-selective adenosine antagonist for adenosinergic receptors, also modulated the expression of the receptors in lung tissue, so that this influence could not be neglected [101,171,172]. Clinically, the spectrum of effects of caffeine on the pulmonary outcome of preterm infants has been demonstrated very well and reflects the experimental findings, as well as the improved respiratory function and reduction in AOP frequency [122].

Schmidt et al. [4] were initially able to demonstrate an improved neuronal outcome, primarily through a reduction in the risk of cerebral palsy, as a result of caffeine in the CAP study. Many clinical studies followed (reported in the introduction) and were supplemented by experimental studies (reported in the results section), thus increasing the robustness of evidence regarding the neuroprotective effect of caffeine. The anti-oxidative, anti-inflammatory, and cell death-reducing effects of caffeine also prevailed here [38,69,109,110,114,115,117,120] and could be displayed more concretely for the protection against oxidative stress for essential processes such as neurogenesis, dendritogenesis, and myelination [38,69,70,71,109,110,112,114,115,117,153]. Improved EEG activity and improved motor function and cognition [116,118,119,121] also suggested a protective effect of caffeine against functional deficits. The influence of caffeine on the antioxidant response, as well as the modulation of adenosine receptors in the brain, indicated similar mechanisms of action for caffeine. Systemically administered caffeine in a single dose or over several days in the vulnerable phase of lung and brain development can reconcile this very well [38,39,68,69,70,71,75].

Is caffeine a ‘game-changer’? In the overarching view, caffeine is the ‘game-changer’ in oxidative stress-relevant damage models and in the clinical setting of premature infants who are exposed to a hyperoxic environment due to premature translation, but also triggered by other unavoidable stressors that cause or intensify oxidative stress [173].

### 4.2. Caffeine, Forever and Ever?

Caffeine is and was used to reduce the frequency of AOP and is the standard drug in the NICU. Side effects such as lower mortality, less BPD, less cerebral palsy, and improved respiratory and motor–cognitive outcomes are well recognized [174]. The use of caffeine therapy is quite variable in the clinic and requires its use in only ventilated premature infants [175], only in premature infants with AOP [176], and, as intensively discussed, with onset within the first three days of life; each case requires different dosing [177]. This variability can certainly also be explained by the increased respiratory instabilities, such as intermittent hypoxia and increased AOP frequency, which correlate with lower gestational age [178]. Currently, caffeine therapy is continued until about 34 corrected weeks of pregnancy (postmenstrual age, PMA) [4,179], while birth before 27 weeks of gestation, BPD, or NEC prolongs the achievement of adequate physiological development [180,181].

Clinical studies on prolonged administration of caffeine are rather scarce. Rhein et al. [182] found that prolonged caffeine therapy for up to 36 weeks of PMA reduced intermittent hypoxia episodes. A continuation of these studies at constant caffeine levels in the therapeutic range demonstrated a reduction in persistent intermittent hypoxia up to 38 weeks PMA [183]. This clinical trial by Rhein et al. was also included in a recent Cochrane analysis [184]. In addition to caffeine therapy in relation to postmenstrual age, two studies were also included that investigated the non-occurrence of symptoms in relation to postmenstrual age. Here, too, early discontinuation of caffeine may be associated with a possible increase in IH episodes [185,186].

Caffeine is a free radical scavenger [37,187], but caffeine also antagonizes cellular adenosine receptors, which are ubiquitously expressed in the brain and respiratory tract [188]. Thus, caffeine also has the potential to trigger effects that could lead to side effects [18,19,20,21,22], such as the induction of a proinflammatory cytokine profile [21] or reduced cerebral oxygenation [19]. When caffeine was administered postnatal to rodents that were in a comparable phase of development to premature infants in the last trimester of pregnancy, adverse effects occurred. In studies of the immature lungs and brains of identical animals, an important finding was induced pro-inflammation [38,68,75], which was often associated with increased cell death rates and induction of the NfκB signaling pathway. Dayanim et al. [108] observed pulmonary infiltration with macrophages and chemokines. Pro-inflammatory responses have also been demonstrated in studies of the developing brain [109,130,132]. These were mostly transient effects, which, in turn, triggered increased oxidative stress [110]. Nevertheless, developmental disabilities in extremely premature newborns are most likely multifactorial but are definitely associated with inflammation at the time of birth. The causal assumption is that transient systemic inflammation can also lead to neurological deficits [189,190,191].

Caffeine, as a single ‘second hit’, has been demonstrated to lead to structural and functional impairments (Table 8, Table 9 and Table 10), such as neuronal disorganization and degeneration [110], reduced vascularization [103], and impaired proliferation and differentiation potential of neuronal progenitor cells [69,70,71,132,134]. Long-term observations then mediated altered behavior and hyperactivity after caffeine treatment [128,135], as well as tachypnea [122]. As expected, caffeine also modulated the expression of adenosine receptors [68,101,110,132]. The involvement of caffeine in the inflammatory response via the adenosine receptors has been demonstrated and also revealed a correlation between caffeine levels and cytokine levels [21,192]. Similarly, caffeine plasma levels in preterm infants are variable but correlate well with the PMA. Due to the immaturity of the hepatic system of preterm infants, caffeine remains almost unmetabolized while renal development continues [193]. In combination with the different expressions as well as the affinity of the adenosine receptors [194] for caffeine, the physiological effects appear to be divergent. Due to the lack of long-term studies in both experimental and clinical settings, it is not possible to provide a comprehensive assessment at this time.

## 5. Conclusions

Caffeine is one of the most commonly used drugs in neonatal intensive care. The benefits of caffeine are evident in terms of reducing apnea of prematurity and minimizing the risk of death, as well as neurological and respiratory comorbidities. In vivo models and premature infant cohorts have demonstrated these effects. These animal studies evaluated caffeine effects in various preterm infant-associated models, including drugs, inflammation, and oxidative stress. Second hits are unavoidable in preterm infants, and the identification of therapeutic strategies to minimize the resulting adverse effects on the immature organism is essential for translation into the clinical setting.

Caffeine therapy is currently administered up to around 34 corrected gestation weeks. The arguments for this are quite sound, as the developmental milestones, such as the maturation of the respiratory system and thus the decrease in the frequency of central apnea, as well as the increase in antioxidant capacity, have been reached. The consideration of prolonging caffeine therapy seems reasonable to increase the response to ongoing oxidative challenges, such as intermittent hypoxia or oxidative stress-inducing triggers, of the preterm infant. Robust clinical preterm and neonatal studies are currently scarce and cannot yet support treatment decisions. However, experimental studies in rodents have shown that caffeine per se as a single ‘second hit’ does not appear to be entirely harmless. In combination with established damage models for oxidative stress, caffeine is a ‘game-changer’ and the caffeine effects can confirm the clinical observations. Of course, the translation into human application must be discussed. The significance seems limited, as, despite comparable ontogenetic phases of neuronal and pulmonary development from rodents to humans, rodents have an already mature antioxidant enzyme system. Nevertheless, it is reasonable to assume that, if the beneficial effects of caffeine under oxygen toxicity are comparable in in vivo models of therapeutic effects in preterm infants, the adverse effects without induced oxidative stress, such as the induction of pro-inflammatory cytokine levels, are also likely to occur. Transient systemic inflammation is a possible predictor of deficits in long-term outcomes. Caffeine therapy of the preterm infant beyond the currently recommended 34 corrected gestation weeks does not appear to be indicated based on current data and requires a high level of further clinical and experimental studies to further ensure the safety of caffeine therapy.

## Figures and Tables

**Figure 1 antioxidants-13-01076-f001:**
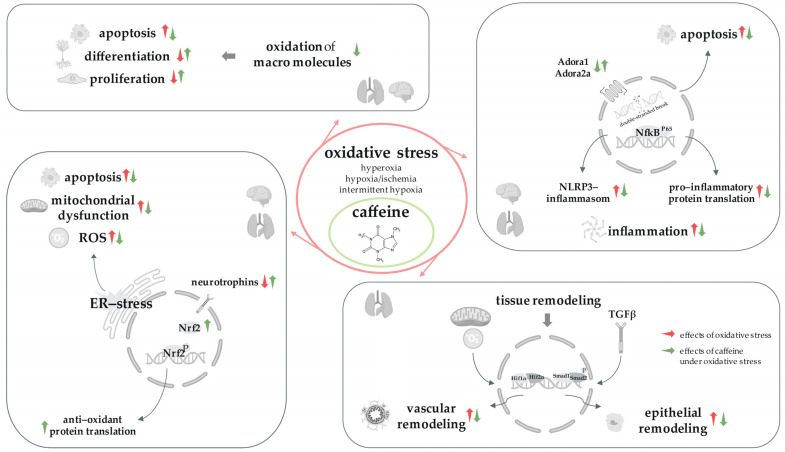
Oxidative stress and caffeine signaling in the brain–lung axis. Oxidative stress-mediated cellular signaling pathways in the immature lung and developing brain, with caffeine as an adequate counteractant to oxygen toxicity.

**Table 1 antioxidants-13-01076-t001:** Caffeine effects in pulmonary hyperoxia-induced damage model.

Species	Oxygen Exposure	Caffeine	Caffeine and Pulmonary Hyperoxia	Ref.
			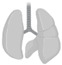	
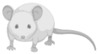 C57Bl/6 mice	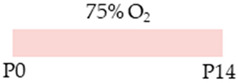	1 g/L lactate damP0–P14	**Pulmonary protection**antioxidativeanti-apoptoticanti-inflammatoryreduced lung architecture simplification increased surfactant protein	**Pulmonary side effects**modulated NfκB signaling pathwaymodulated AdoRA2a expression**Side effects**weight gain	[101]
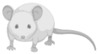 C57Bl/6 mice	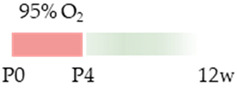	20 mg/kg i.p.P0–P4	**Pulmonary protection**improved microvascularization (male)reduced lung architecture simplification	**Pulmonary side effects**modulated HIF signaling pathway**Side effects**weight gain	[102]
 SD rat	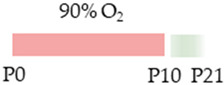	20 mg/kg i.p.P2–P10/P21	**Pulmonary protection**antioxidativeanti-apoptoticimproved microvascularization reduced lung architecture simplification reduced endoplasmic (ER) stressreduced mitochondrial dysfunction		[103]
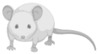 C57Bl/6 mice	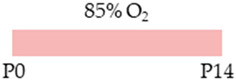	25 mg/kg i.p.P0–P13	**Pulmonary protection**none	**Pulmonary side effects**increased phosphorylation Smad2**Side effects**weight gain	[104]
 SD rat	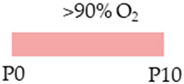	20 mg/kg i.p.P2–P21	**Pulmonary protection**improved microvascularization reduced lung architecture simplification	**Pulmonary side effects**reduced cAMP level **Side effects**reduced mortalityweight gain	[105]
 SD rat	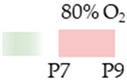	10 mg/kg i.p.P7	**Pulmonary protection**antioxidantanti-apoptoticanti-inflammatoryreduced lung architecture simplification		[106]
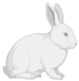 New Zealand white rabbit	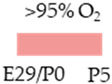	10 mg/kg i.p.P05 mg/kg i.p.P1–P5	**Pulmonary protection**anti-inflammatoryimproved lung functionreduced lung architecture simplification		[107]
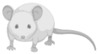 FVB/n mice	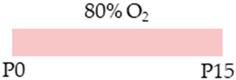	20 mg/kg i.p.P110 mg/kg i.p.P2–P15	**Pulmonary protection**none	**Pulmonary damage**pro-apoptoticpro-inflammatoryexacerbated lung architecturereduced AECII counts**Pulmonary side effects**reduced AdoRA2a expression	[108]

The colors in the schematic experimental designs indicate hyperoxia (red) with increasing intensity to inspiratory oxygen concentration and exposure to room air (21% O_2_, green).

**Table 2 antioxidants-13-01076-t002:** Caffeine effects in neuronal hyperoxia-induced damage model.

Species	Oxygen Exposure	Caffeine	Caffeine and Neuronal Hyperoxia	Ref.
			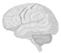	
 SD rat	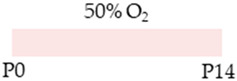	20 mg/kg i.p. P05 mg/kg i.p.P1–P14	**Neuroprotection**anti-apoptoticanti-inflammatoryprotected against demyelination	**Neurodegeneration/side effects** n/a	[109]
 SD rat	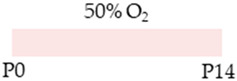	LoC:20 mg/kg i.p. P05 mg/kg i.p. P1–P14HiC:80 mg/kg i.p. P020 mg/kg i.p. P1–P14	**Neuroprotection**LoC: antioxidantanti-apoptotic reduced neuronal pyknosisincreased neurons and dendritogenesisHiC:antioxidantanti-apoptotic increased neurons and dendritogenesis protected from demyelination	**Neurodegeneration/side effects**LoC:reduced AdoRA1a/2a expressionweight gainHiC:intraventricular haemorrhageshistopathological abnormalitiesreduced AdoRA1a/2a expression weight loss	[110]

The red color in the schematic experimental designs indicates hyperoxia (red) in the inspiratory oxygen concentration.

**Table 3 antioxidants-13-01076-t003:** Caffeine effects in pulmonary and neuronal hyperoxia-induced damage model.

Species	Oxygen Exposure	Caffeine	Caffeine and Pulmonary-Neuronal Hyperoxia	Ref.
			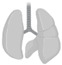	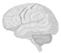	
 Wistar rat	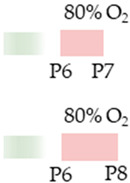	10 mg/kg i.p. P610 mg/kg i.p. P6	**Pulmonary protection**anti-inflammatoryreduced lung architecture simplification	**Neuroprotection**antioxidantanti-apoptoticanti-inflammatoryprotected from tissue remodelingprotected from impaired hippocampal neurogenesis**Neuronal side effects**modulated NfκB/Nrf2 signaling pathway	[38,69,75]The studies use the identical cohorts.
 Wistar rat	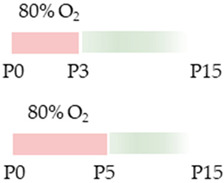	10 mg/kg i.p. P0–P210 mg/kg i.p. P0–P4	**Pulmonale Protection**antioxidantanti-apoptoticanti-inflammatory**Pulmonary side effects**modulated NfκB/Nrf2 signaling pathwaymodulated AdoRA1 expression	**Neuroprotection**protected from impaired hippocampal neurogenesis protected from impaired cerebellar neurogenesis and migrationrescued neurotrophins	[39,68,70,71]The studies use the identical cohorts.

The colors in the schematic experimental designs indicate hyperoxia (red) in the inspiratory oxygen concentration and exposure to room air (21% O_2_, green).

**Table 4 antioxidants-13-01076-t004:** Caffeine effects in neuronal hypoxia-induced damage model.

Species	Oxygen Exposure	Caffeine	Caffeine and Neuronal Hypoxia	Ref.
			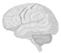	
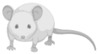 mice	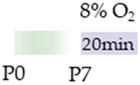	20 mg/kg p.o. P415 mg/kg p.o.P5–P7	**Neuroprotection**reduced HIF1α cortical accumulation	**Neurodegeneration/side effects** n/a	[111]
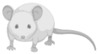 C57Bl/6 mice	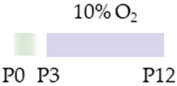	0.3 g/L lactate damP2–P12	**Neuroprotection**reduced ventriculomegaly protected against demyelination	**Neurodegeneration/side effects** n/a	[112]

The colors in the schematic experimental designs indicate hypoxia (purple) in the inspiratory oxygen concentration and exposure to room air (21% O_2_, green).

**Table 5 antioxidants-13-01076-t005:** Caffeine effects in neuronal hypoxia–ischemia-induced damage model.

Species	Oxygen Exposure	Caffeine	Caffeine and Neuronal Hypoxia-Ischemia	Ref.
			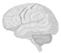	
 SD rat	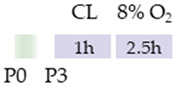	10 mg/kg i.p. P2–P6before HIandpost HI	**Neuroprotection**anti-apoptotic	**Neuronal side effects**improved mitochondrial functionpromoted effects of deacetylation	[113]
 SD rat	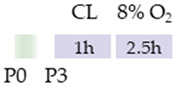	10 mg/kg i.p. P2–P6before HIandpost HI	**Neuroprotection**anti-inflammatoryrestored reduced synapses proteinsalleviated the suppressed cognitive impairmentreduced ventriculomegaly protected against demyelinationpromote microglial polarization to the M2 phenotype	**Neurodegeneration/side effects** n/a	[114,115]The studies use the identical cohorts.
 SD rat	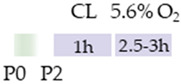	0.3 g/L lactate damE8–P16before HIandP3–P16post HI	**Neuroprotection**before HI: reduced number of brain infarctsreduced infarct volumerestored suppressed EEG brain activitypost HI: restored suppressed EEG brain activity	**Neurodegeneration/side effects** n/a	[116]
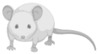 C57Bl/6 mice	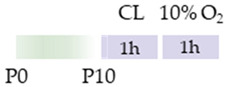	5 mg/kg i.p. 0 h, 6 h, 12 h, and 24 hpost HI	**Neuroprotection**anti-apoptoticanti-inflammatoryreduced infarct volumeprotected against demyelinationreduced microglia-activation	**Neurodegeneration/side effects** n/a	[117]
 Wistar rat	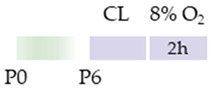	10 mg/kg i.p. 0 h, 24 hpost HI	**Neuroprotection**improved cognitive and motoric learning (male)	**Neurodegeneration/side effects** n/a	[118]
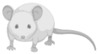 C57Bl/6 mice	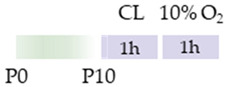	5 mg/kg i.p. 0 hpost HI	**Neuroprotection**improved motoric learning	**Neurodegeneration/side effects** n/a	[119]
 Wistar rat	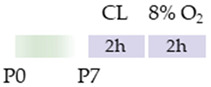	10 mg/kg i.p. before HI and0 h, 24 h, 48 h, and 72 hpost HI	**Neuroprotection**anti-apoptotic (HC, cortex)	**Neurodegeneration/side effects** n/a	[120]
 Wistar rat	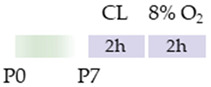	10 mg/kg i.p. 0 h post HI	**Neuroprotection**improved spatial learning (male)	**Neurodegeneration/side effects** n/a	[121]

The colors in the schematic experimental designs indicate the carotid artery ligation (CL) and hypoxia (purple) in the inspiratory oxygen concentration and exposure to room air (21% O_2_, green).

**Table 6 antioxidants-13-01076-t006:** Caffeine effects in pulmonal intermittent hypoxia-induced damage model.

Species	Oxygen Exposure	Caffeine	Caffeine and Pulmonal Intermittent Hypoxia	Ref.
			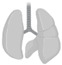	
 SD rat	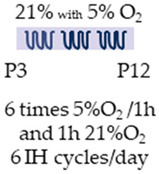	15 mg/kg p.o. P3–P15 *10 mg/kg i.p.P3–P15 **	**Pulmonary protection**reduced frequency of apnea	**Pulmonary side effects**n/a	[122] *[123] **

The colors in the schematic experimental designs show the hypoxia (purple) in the inspiratory oxygen concentration, with waves to represent the intermittent oxygen concentrations. * The indicated concentration refers to [122]. ** The indicated concentration refers to [123].

**Table 7 antioxidants-13-01076-t007:** Caffeine effects in neuronal intermittent hypoxia-induced damage model.

Species	Oxygen Exposure	Caffeine	Caffeine and Neuronal Intermittent Hypoxia	Ref.
			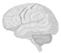	
 SD rat	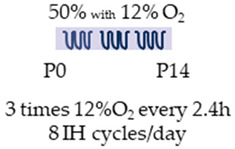	20 mg/kg i.p. P05 mg/kg i.p.P1–P14	**Neuroprotection**anti-inflammatoryreduced histopathological abnormalities	**Neuronal side effects**hypermyelination**Side effects**reduced enhanced brain/body weight ratio	[109]
 SD rat	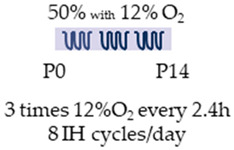	LoC:20 mg/kg i.p. P05 mg/kg i.p. P1–P14HiC:80 mg/kg i.p. P020 mg/kg i.p. P1–P14	**Neuroprotection/side effects**LoC: anti-apoptotic increased neurons and dendritogenesisHiC:antioxidantanti-apoptotic increased neurons and dendritogenesis	**Neurodegeneration/side effects**LoC:pro-oxidantreinforced demyelinationhistopathological abnormalitiesNeuronal side effectsLoC: reduced AdoRA1a expressioninduced AdoRA2a expressionHiC:reduced AdoRA1a expressionSide effectsLoC:induced body lengthHiC:reduced brain weightweight loss	[110]

The color in the schematic experimental designs shows the hypoxia (purple) in the inspiratory oxygen concentration, with waves to represent the intermittent oxygen concentrations.

**Table 8 antioxidants-13-01076-t008:** Caffeine effects in the immature lung under standard conditions.

Species	Oxygen Exposure	Caffeine	Caffeine and the Immature Lung	Ref.
			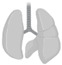	
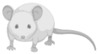 C57Bl/6 mice	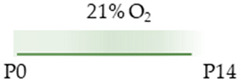	1 g/L lactate damP0–P14	**Pulmonary damage**none	**Pulmonary side effects**reduced AdoRA2a expression	[101]
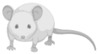 C57Bl/6 mice	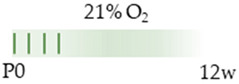	20 mg/kg i.p. P0–P4	**Pulmonary damage**none	**Pulmonary side effects**induced HIF mapped transcriptsinduced vascular-associated transcriptsweight gain (male)	[102]
 SD rat	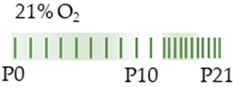	20 mg/kg i.p.P2–P10/P21	**Pulmonary damage**increased ER-phagy associated proteinactivated UPR (linked to fibrosis and inflammation)decreased blood vessel formation	**Pulmonary side effects** n/a	[103]
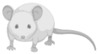 C57Bl/6 mice	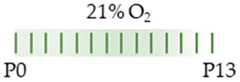	25 mg/kg i.p. P0–P13	**Pulmonary damage**decreased phosphorylation Smad2 (linked to epithelial remodeling)	**Pulmonary side effects** n/a	[104]
 SD rat	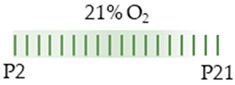	20 mg/kg i.p. P2–P21	**Pulmonary damage** n/a	**Side effects**weight loss	[105]
 SD rat	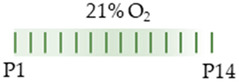	10 mg/kg p.o. P1–P14	**Pulmonary damage** n/a	**Pulmonary side effects** n/a	[124]
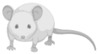 FVB/n mice	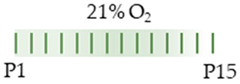	20 mg/kg i.p. P110 mg/kg i.p. P2–P15	**Pulmonary damage**pro-apoptoticpro-inflammatory with MΦ infiltration and chemokine-inductionreduced surfactant protein C transcriptreduced AEC II	**Pulmonary side effects** n/a	[108]
 SD rat	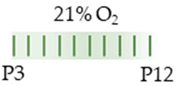	15 mg/kg p.o.P3–P12	**Pulmonary damage** n/a	**Pulmonary side effects**increased minute ventilationincreased breathing frequency	[122]

The color in the schematic experimental designs shows normoxia (21% O_2_, green) with a horizontal line during sustained caffeine administration or in vertical lines to indicate the frequency of caffeine administration.

**Table 9 antioxidants-13-01076-t009:** Caffeine effects in the developing brain under standard conditions.

Species	Oxygen Exposure	Caffeine	Caffeine and the Developing Bain	Ref.
			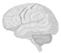	
 SD rat	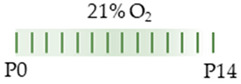	20 mg/kg i.p. P05 mg/kg i.p.P1–P14	**Neurodegeneration**pro-inflammatory	**Neuronal side effects**hypermyelination	[109]
 SD rat	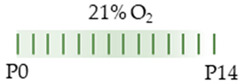	LoC:20 mg/kg i.p. P05 mg/kg i.p. P1–P14HiC:80 mg/kg i.p. P020 mg/kg i.p. P1–P14	**Neurodegeneration**LoC: pro-oxidantneuronal disorganizationHiC:pro-oxidantpro-apoptotic pro-necrotic neuronal degeneration and disorganization	**Neuronal side effects**LoC: hypermyelinationincreased neurons per area and dendrites per neuronreduced AdoRA1 and AdoRA2a expressionHiC:hypermyelinationincreased neurons per area and dendrites per neuronreduced AdoRA1 and AdoRA2a expression	[110]
 Wistar rat	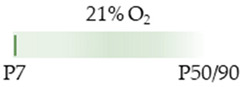	10 mg/kg s.c. P7	**Neurodevelopment**hyperlocomotionreduced social interactionreduced contextual fear conditioning	**Neuronal side effects** n/a	[128]
 SD rat	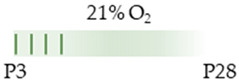	50 mg/kg s.c. P3–P6	**Neurodegeneration**pro-apoptotic impaired mitochondrial dysfunction	**Neuronal side effects** n/a	[129]
 Wistar rat	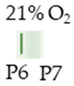	10 mg/kg i.p. P6	**Neurodegeneration**reduced hippocampal proliferation capacityreduced differentiation potential in hippocampal neural linage	**Neuronal side effects**increased neurotrophin expression	[130]
 Long Evans rat	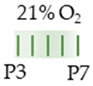	20 mg/kg s.c.P315 mg/kg s.c.P4–P7	**Neurodegeneration**reduced neurons (Hth and CA1)pro-apoptotic (CA1, Hth, Th, WM)	**Neuronal side effects** n/a	[131]
 Wistar rat	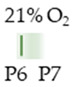	10 mg/kg i.p. P6	**Neurodegeneration**pro-apoptotic (DG, Hth)pro-inflammatory	**Neuronal side effects**increased AdoRA1 and AdoRA2a expression	[132]
 SD rat	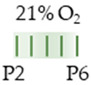	20 mg/kg i.p. P215 mg/kg i.p.P3–P6	**Neurodegeneration**none	**Neuronal side effects**induced early maturation adenosinergic system (Hth, brain stem)induced AdoRA1 and AdoRA2a expression	[133]
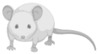 mice	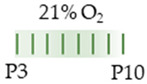	10 mg/kg i.p. P32.5 mg/kg i.p.P4–P10	**Neurodegeneration**reduced proliferation SVZ and WMreduced astrocytogenesis cortex and WM	**Neuronal side effects** n/a	[134]
 Wistar rat	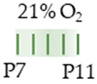	10 mg/kg s.c.P7–P1120 mg/kg s.c.P7–P11	**Neurodevelopment**reduced motoric performancehyperactivity	**Neuronal side effects** n/a	[135]

The color in the schematic experimental designs shows normoxia (21% O_2_, green) with vertical lines to indicate the frequency of caffeine administration.

**Table 10 antioxidants-13-01076-t010:** Caffeine effects in the immature lung and the developing brain under standard conditions.

Species	Oxygen Exposure	Caffeine	Caffeine and the Immature Lung and the Developing Brain	Ref.
			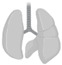	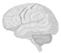	
 Wistar rat	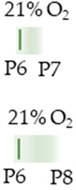	10 mg/kg i.p. P610 mg/kg i.p. P6	**Pulmonary damage**pro-inflammatory	**Neurodegeneration**pro-inflammatorymodulated NfκB signaling pathwayreduced hippocampal proliferationreduced hippocampal neurogenesisinhibited physiological apoptosis	[38,69,75]The studies use the identical cohorts.
 Wistar rat	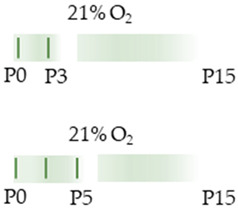	10 mg/kg i.p. P0–P210 mg/kg i.p. P0–P4	**Pulmonary damage**pro-inflammatory**Pulmonary side effects**modulated NfκB/Nrf2 signaling pathway and AdoRa1 expression **Side effects**gain loss	**Neurodegeneration**reduced hippocampal neurogenesisreduced hippocampal proliferationreduced neurotrophin expression**Neuronal side effects**increased AdoRA1, AdoRA2a, and AdoRA2b expressionmodulated cerebellar neurogenesis and migrationinduced cerebellar proliferation	[39,68,70,71]The studies use the identical cohorts.

The color in the schematic experimental designs shows normoxia (21% O_2_, green) with vertical lines to indicate the frequency of caffeine administration.

## Data Availability

The data presented in this study are available in the tables of this article.

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
