# Peer review of "Caffeine: The Story beyond Oxygen-Induced Lung and Brain Injury in Neonatal Animal Models—A Narrative Review"

_antioxidants, 2024, doi:10.3390/antiox13091076_

Round 1

Reviewer 1 Report

I have read this paper with a background on perinatal clinical research and clinical pharmacology. I value the effort, that is narrative in its approach, but has applied a structured search.

Some reflections

Are you ‘sure’ on the causal iink between oxidative stress and asthma In former preterms ? if so, please support this with evidence.

There is (really just) a published Cochrane review on strategies for cessation of caffeine administration in preterm infants (PMID 39045901). I’m not involved in this paper, so that I can recommend to include this, as it covers the topic that you address from a preclinical point of view.

Last alinea (from Line 45). Please make it clearer that these are findings in human preterms (or at least, this is my assessment).

On 1.2, I dare to suggest some adaptations.

‘due to the reduced proportion of CYP1A2’ = perhaps due to lower, ontogeny related CYP1A2 activity’

I have not checked again, but I understood that the switch from ‘renal’ to ‘hepatic’ rather occurs at 3-4 months (post term age). Finally, as CYP1A2 is not yet relevant, the polymorphisms are not yet relevant in neonates and young infants.

Related to 1.3. I would suggest to add a refer to the BOOST trials (saturation/outcomes studies) to put things into perspective.

Line 142, Ref 76: is this really causal, or rather an association (BPD causing neurodevelopmental issues ?)

Textual

You commonly use ‘injured’ models in your subtitles, I think that injury is perhaps more accurate.

Line 194: suggest to add this recommended corrected gestational age (you mention this lower)

Reviewer 2 Report

The paper entitled: "Caffeine: The Story Beyond Oxygen-induced Lung and Brain Injury in Neonatal Animal Models" is very well written and without a doubt will be interesting for both clinicians and researchers. It describes caffeine effects using various injury small animal models including preterm and oxidative stress-adapted rodent models. I would note, that even though laboratory rodents are commonly used to study brain development,  the mouse brain is underdeveloped at birth contrasting to human newborns, thus making these small animal models less optimal for the research of the brain development of human infants. Recently, however, the problem has been alleviated by the use of large animal models, whose brain developmental programs are particularly similar to those in humans.  Adding a paragraph regarding the benefits of the use of large animal models in such studies would greatly improve the paper (examples: PMCID: PMC9744950; PMCID: PMC9913628; PMCID: PMC7464467; PMCID: PMC6291230).

Minor: I suggest increasing the resolution of Fig1.

I suggest increasing the resolution of Fig1 (line490).

Round 2

Reviewer 2 Report

This paper is very well written. I recommend to accept it in its present form.

None